# Dereplication by High-Performance Liquid Chromatography (HPLC) with Quadrupole-Time-of-Flight Mass Spectroscopy (qTOF-MS) and Antiviral Activities of Phlorotannins from *Ecklonia cava*

**DOI:** 10.3390/md17030149

**Published:** 2019-03-04

**Authors:** Hyo Moon Cho, Thi Phuong Doan, Thi Kim Quy Ha, Hyun Woo Kim, Ba Wool Lee, Ha Thanh Tung Pham, Tae Oh Cho, Won Keun Oh

**Affiliations:** 1Korea Bioactive Natural Material Bank, Research Institute of Pharmaceutical Sciences, College of Pharmacy, Seoul National University, Seoul 08826, Korea; chgyans@naver.com (H.M.C.); phuongdoan@snu.ac.kr (T.P.D.); htkquy@ctu.edu.vn (T.K.Q.H.); kimkami2@snu.ac.kr (H.W.K.); paul36@snu.ac.kr (B.W.L.); thtungdl@gmail.com (H.T.T.P.); 2Marine Bio Research Center, Department of Life Science, Chosun University, Gwangju 501-759, Korea; tocho@chosun.ac.kr

**Keywords:** *Ecklonia cava*, bioactivity-guided isolation, dereplication, relative mass defect, HPLC-qTOFMS, antiviral, phlorotannins

## Abstract

*Ecklonia cava* is edible seaweed that is found in Asian countries, such as Japan and Korea; and, its major components include fucoidan and phlorotannins. Phlorotannins that are isolated from *E. cava* are well-known to have an antioxidant effect and strong antiviral activity against porcine epidemic diarrhea virus (PEDV), which has a high mortality rate in piglets. In this study, the bioactive components were determined based on two different approaches: (i) bio-guided isolation using the antiviral activity against the H1N1 viral strain, which is a representative influenza virus that originates from swine and (ii) high-resolution mass spectrometry-based dereplication, including relative mass defects (RMDs) and HPLC-qTOFMS fragmentation analysis. The EC70 fraction showed the strongest antiviral activity and contained thirteen phlorotannins, which were predicted by dereplication. Ten compounds were directly isolated from *E. cava* extract and then identified. Moreover, the dereplication method allowed for the discovery of two new phlorotannins. The structures of these two isolated compounds were elucidated using NMR techniques and HPLC-qTOFMS fragmentation analysis. In addition, molecular modelling was applied to determine the absolute configurations of the two new compounds. The antiviral activities of seven major phlorotannins in active fraction were evaluated against two influenza A viral strains (H1N1 and H9N2). Six of the compounds showed moderate to strong effects on both of the viruses and phlorofucofuroeckol A (**12**), which showed an EC_50_ value of 13.48 ± 1.93 μM, is a potential active antiviral component of *E. cava*.

## 1. Introduction

*Ecklonia cava*, which is edible seaweed belonging to the Laminariaceae family, is found in Asian countries, such as Japan and Korea. *E. cava* is a nutrient-dense food and it contains many types of organic compounds, including fucoidan and phlorotannins [1], which have various biological activities, such as antioxidant [2,3], anti-allergic [4], anti-plasmin inhibitory [5], anticancer [6], antihypertensive [7], and elastase and hyaluronidase inhibitory effects [8], as well as strong antiviral activity against porcine epidemic diarrhea virus (PEDV) (a coronavirus) [9]. 

Influenza, similar to coronaviruses, is also a contagious viral infection that causes acute respiratory illnesses and rapidly spreads through outbreaks [10,11]. The influenza virus belongs to the Orthomyxoviridae family, and there are three different types: A, B, and C. Among them, types A and B are of great concern as human pathogens and result in seasonal or interpandemic epidemics as well as global pandemics, such as those that are caused by influenza A viruses [12,13]. Over the past 100 years, four pandemics have been reported: H1N1 Spanish influenza (1918), H2N2 Asian influenza (1957), H3N2 Hong Kong influenza (1968), and recently, 2009 H1N1 (H1N1pdm09), which are all caused by influenza A viruses [12,14]. In each pandemic, the number of novel virus strains arose and spread through human communities, leading to the substantial morbidity and mortality related to bacterial pneumonia [11,15]. The influenza A viruses are single-stranded negative-sense RNA viruses that are subtyped on the basis of the composition of their two surface glycoproteins: hemagglutinin (HA) and neuraminidase (NA) [16]. The genome of influenza A viruses comprises 8 separated gene segments encoding 16 proteins and is surrounded by a liquid envelope [17,18]. Two glycoproteins, HA and NA, are found in the viral envelopes, and these are considered to be the basis of antigenicity. HA initiates the infection of influenza virus by binding to the α-2,6-linked sialic acid and/or α-2,3-linked sialic acid receptors on the surface of the host cell, followed by receptor-mediated endocytosis of the virion into the cell [17,19]. The NA protein serves as a sialidase, and it cleaves the link between the sialic acid and the HA protein to release the virus particles [11,19]. To date, the Food and Drug Administration (FDA) [20] have approved only three drugs: Tamiflu (oseltamivir phosphate), Relenza (zanamivir), and Rapivab (peramivir).

Dereplication is the term that is used to discover new natural materials through the rapid identification of compounds that are based on new analytical tools, such as MS and NMR spectroscopy, to avoid the re-isolation of known components. In particular, the rapid development of modern MS techniques has accelerated dereplication strategies using high-performance liquid chromatography—quadrupole time-of-flight mass (HPLC-qTOFMS) spectra with tandem database searching to identify the known compounds in the extract [21]. By increasing the use of high-resolution mass spectrometry (HRMS) for measuring exact masses, the concept of “mass defect” is also increasingly being used in natural products research [22]. The mass defect is calculated as the deviation between the exact mass and the nominal mass (integer-rounded mass) of a compound [22,23]. Several methods have been developed to simplify the use of mass defects, including Kendrick mass defects, mass defect filtering, and fractional mass, which have played an important role in identifying many previously unknown compounds. In addition, a relative mass defect (RMD), calculated as (mass defect/measured monoisotopic mass) × 10^6^ in ppm, was recently introduced [23,24]. The RMD remains constant for classes that share the same fractional hydrogen content, which is useful in classifying the compounds into groups with the same skeleton [23]. While fatty acids or hydrocarbons with high hydrogen contents show high RMDs (400 to 600 ppm), the values for polyphenolic metabolites are often less than 300 ppm due to their high oxygen contents [23]. In addition to modern techniques, such as HPLC-qTOFMS, classical bio-guided isolation has played an important role in the primary bioactive screening and identification of potential natural products. Pre-fractionation has recently been applied in bioactivity-guided isolations as an important step in obtaining bioactive materials. This is because the activities of minor components can be masked or decreased by more abundant inactive compounds in the total extract [25]. The combination of bio-guided isolation and dereplication enables the rapid and effective identification of the potential bioactive compounds [26].

The aim of the present study was to develop a new antiviral agent from *E. cava*. Specifically, we want to develop dereplication methods, including classification that is based on RMD and HPLC-qTOFMS, to accurately and rapidly identify bioactive phlorotannin-type compounds in *E. cava*, as well as to easily target new compounds. 

## 2. Results and Discussion

### 2.1. Bioactivity-Guided Isolation and Dereplication of the Active Antiviral Fraction

#### 2.1.1. Bioactivity-Guided Fractionation Based on Antiviral Activity Against Human Influenza H1N1 A/PR/8/34 Virus 

The 80% MeOH extract of *E. cava*, which showed antiviral activity at 10 µg/mL in a cytopathic effect (CPE) assay when compared to ribavirin (10 µM) as a positive control (Figure 1A), was divided into five fractions on a Diaion HP-20 column, with a gradient of MeOH and H_2_O (0/100→100/0, *v*/*v*), and finally acetone. The antiviral activities of the five fractions indicated that the sub-fraction eluted with 70% MeOH (EC70) had the strongest antiviral activity against H1N1-infected cells at three different concentrations (5, 10, and 20 µg/mL, Figure 1B). Thus, the EC70 fraction was chosen for further bio-guided isolation and dereplication using an HPLC-qTOFMS to identify the predicted bioactive compounds (Figure 2). Potential components against H1N1 and H9N2 viruses were further tested against viral protein synthesis of the H1N1 A/PR/8/34 virus (Figure 3).

#### 2.1.2. Dereplication and Isolation of the Phlorotannins from the EC70 Fraction Using HPLC-qTOFMS and RMD 

The dereplication of the secondary metabolites from *E. cava* was accomplished by HPLC-qTOFMS analysis of the active fraction (EC70). The HPLC-qTOFMS data were used for database construction based on the chromatographic retention times, full-scan (high-resolution) mass spectra, and MS/MS spectra in negative electrospray ionization (ESI) mode. The secondary metabolites in the EC70 fraction were identified by comparison with compounds that were previously reported in the *Ecklonia* genus (12 species listed in Appendix A) in online (Scifinder) and in-house databases that are based on exact mass and RMD using MassHunter quantitative B4.00 software. The identified compounds are shown as the base peaks of the MS and MS/MS chromatograms in Figure 4B–E. The MS^1^ and MS^2^ data of all the peaks that were detected in the spectra are listed in Table 1. The RMD value of each peak was used to determine its chemical skeleton. The structures of the components, which were identified by the comparison of the MS/MS fragmentations of the peaks in Table 1 to those in the in-house library, are shown in Figure 2. 

The components on the EC70 fraction could be classified as phlorotannins, fatty acids, and others compounds with different ranges of RMD values. The RMD values of the detected phlorotannins were smaller than those of the other compounds, which are only approximately 10 to 200 ppm due to the nature of the structure of phlorotannins (Appendix A). Positive absolute mass defects are usually observed in compounds that contain a large number of hydrogen atoms (mass defect +7.83 mDa), whereas the lower values are observed in compounds with large numbers of oxygen (mass defect −5.09 mDa) and nitrogen atoms (mass defect +3.07 mDa) [23]. Fatty acids contain many hydrogens and they have large RMD values (as suggested by compounds **21**, **22**, **24**, and **26**–**29**, Table 1 and Figure 2), but oxygen is a major elementary building block of phlorotannins, which reduces the RMD values of this group, particularly in this study, to less than 200 ppm (as demonstrated by compounds **6**–**18**, Table 1). These features are very helpful for further studying the genus *Ecklonia,* which contains phlorotannins as major components. As the first step of screening by HPLC-qTOFMS, the calculation of RMD values for all detected peaks facilitates the identification of phlorotannins and allows researchers to dereplicate the reported compounds, especially in this genus. The RMD values in the list of unidentified peaks in Table 1 are also useful for estimating the types of structures that are associated with detected peaks.

Furthermore, polyphenols have lower hydrogen contents and higher oxygen contents when compared to other types of compounds, resulting in lowered RMD values, which is consistent with what is seen in phlorotannins. Thus, in further studies, classification that is based on the RMD value is suitable for other types of polyphenolic compounds and it can be broadly applied in the screening of natural resources. In particular, for most species of brown algae, phlorotannin-type compounds can easily be classified because they are the main components of this species.

Phlorotannins, which are composed of phloroglucinol (1,3,5-trihydroxybenzene) units [27,28], are well-known as major components of the genus *Ecklonia* [9]. Overall, phlorotannins are divided into four classes that are based on the type of linkage that connects the phloroglucinol units, including ether-bound phlorotannins (fuhalols and phlorethols), phenyl-bound phlorotannins (fucols), ether and phenyl-bound phlorotannins (fucophlorethols), and dibenzo-dioxin-bound fucophlorethols (eckols and carmalols) [27,28]. Monomers in each class can be linked at various positions; thus, it is possible to have many structural isomers with the same molecular weight [27,28]. The fragmentation patterns that were observed for these peaks are therefore key to structural identification, as they indicate different losses of phloroglucinol monomers and hydroxyl groups [27]. In most previous studies, it has been necessary to isolate the compounds and use a combination of MS and NMR spectroscopy to elucidate the exact structures of phlorotannins [27]. Recently, an attempt was made to classify the types of phlorotannins in *Sargassum fursiforme* without isolation using ultrahigh-performance liquid chromatography triple quadrupole tandem mass spectrometry (UHPLC-QqQ-MS) for the first time. In this study, we proposed an efficient method to dereplicate the various phlorotannins derivatives, which are major constituents in seaweeds, based on preliminary HPLC-qTOFMS accompanied by RMD analysis. Moreover, the hypothesis of dereplication was further validated by the isolation and structural elucidation of phlorotannins using NMR and HPLC-qTOFMS. The MS/MS data of the peaks that were detected in the EC70 fraction and the anticipated components are listed in Table 1. As shown in Figure 4C and Table 1, forty-one peaks were observed in the HPLC-qTOFMS spectroscopic data of the active fraction, and compounds **6**–**18** (Table 1) were identified as phlorotannins based on a comparison of their RMD values, molecular weights, and MS/MS fragments to the literature data. According to their MS/MS data, thirteen compounds were identified as eckol-type phlorotannins, which contain at least one 1,4-dibenzodioxin fragment in their structures and have been mostly found in *Ecklonia* and some other genus of seaweeds [27,28]. The fragment peaks that were observed in the MS/MS patterns of these compounds revealed the losses of phloroglucinol (126 amu), pentahydroxyl dibenzodioxin (263 amu) and eckol (371 amu) based on the dominant product ions at *m/z* 229 amu [371 (eckol) −126 (phloroglucinol) −16 (oxygen atom) amu] and 261 amu [263 (pentahydroxyl dibenzodioxin) −2H]. For example, bieckol (compound **15**, *m/z* 741.0737, Table 1) showed product ions at *m/z* 615, 479, 371, and 260 from losses of one phloroglucinol (−126 amu), two phloroglucinols and a water (−249 −18 amu), eckol (−371 amu), and eckol with a phloroglucinol being followed by the attachment of an oxygen atom (−371 –126 +16 amu).

According to dereplication results, ten known phlorotannins (**6**, **7**, **10**–**16**, and **18**) were directly isolated from the EC70 fraction to validate the accuracy of the identification. These ten compounds were measured HPLC-qTOFMS in negative ion mode at a collision energy of 50 eV to further confirm their retention times (Appendix A). The fragmentation patterns of the isolated phlorotannins are presented in Appendix A. The results revealed that HPLC-qTOFMS was a powerful tool for identifying the structure of phlorotannins, especially in the cases of isomers, as it can provide sufficient data for the elucidation of the chemical structures. As an example, the MS/MS fragmentation pattern of compound **18** (*m/z* 974.1038) is presented in detail. As identified in Table 1, compound **18** was established as compound 974-A due to its low RMD value (107 ppm) and the characteristic fragments that were observed at *m/z* 973, 707, 353, and 229 [29]. After isolation, the exact structure of **18** was further confirmed by NMR spectroscopy in conjunction with MS/MS fragmentation (Appendix A). The product ions of **18** are shown in Appendix A and they are due to the losses of bifuhalol (–266 amu), three phlorotannins and one water (−375 −18 amu), and four phlorotannin units and one water (−448 −18 amu). Similarly, although compounds **10**, **11**, and **15** showed the same molecular ions ([M − H]^−^ of 741.0647, 741.0723, and 741.0737, respectively) (Table 1), they could be distinguished by their NMR data and different fragment ions in their MS/MS patterns (Appendix A). In particular, compound **10** showed fragments at *m/z* 619.0680, 495.0568, and 250.0477 from the losses of phloroglucinol (126 amu) and two equivalents of phloroglucinol (250 amu). Compound **11** generated product ion peaks at *m/z* 565.3355, 437.0064, 369.0253, and 230.9843 from the losses of phloroglucinol and three waters (176 amu) and a dioxinodehydroeckol (369 amu). **15** showed product ions at *m/z* 477.0398, 369.0198, 351.0087, and 125.0224, due to the losses of two molecules of phloroglucinol and water (264 amu) and three phloroglucinol units (372 amu). 

#### 2.1.3. Antiviral Activities of the Major Phlorotannins in the EC70 Active Fraction against H1N1 and H9N2

In this study, seven major phlorotannins that were isolated from the EC70 fraction using bioactivity-guided isolation were tested for their antiviral activities against two influenza A viral strains (H1N1 A/PR/8/34 and H9N2 virus strains) (Figure 1C,D). The results indicated that six compounds (**6** and **10**–**14**) had moderate to strong antiviral activities against both viral strains at a concentration of 20 µM. Compounds **11**–**14**, which displayed potent antiviral effects, were further tested against viral protein synthesis of the H1N1 A/PR/8/34 virus and compared to ribavirin as a positive control. The results, as shown in Figure 3A and Appendix A, suggested that compound **12** was the most active compound with an EC_50_ of 13.48 ± 1.93 μM. Thus, the dose-dependent inhibitory effects of compound **12** on viral protein synthesis were evaluated at four different concentrations 5, 10, 20, and 40 μM in Madin–Darby canine kidney (MDCK) cells. As shown in Figure 3B, it is clear that, at higher concentrations, **12** more effectively inhibited protein expression in viral-infected cells. Compound **12** decreased the expression of neuraminidase and hemagglutinin at 10 μM, and the strongest inhibition capacity was observed at 40 µM. The fluorescence images that are shown in Figure 3C also explain the inhibitory effect of compound **12** on neuraminidase protein expression in H1N1 infected MDCK cells. The results suggested that compound **12** showed strong antiviral activity through inhibiting the expression of surface glycoproteins, hemagglutinin, and neuraminidase.

### 2.2. Identification of Two New Phlorotannins from the EC30 Fraction of E. cava using Dereplication and RMD Rules

The results from the dereplication of the EC70 fraction suggested a quick method to identify phlorotannins. Hence, we attempted to isolate the new phlorotannins from the other fractions of *E. cava* based on their MS/MS fragmentation patterns and RMD values. The HPLC-qTOFMS data from fraction EC30 of *E. cava* was exported and the RMD values of the detected peaks were calculated. Compounds **1** (*m/z* 743.0881, Table 1) and **2** (*m/z* 1113.1146, Table 1) showed RMD values that were between 0 and 200 ppm, and their fragments, which were not presented in the in-house database or online database of the *Ecklonia* genus, suggested that these compounds might be new phlorotannins (Figure 4A). Thus, the two compounds were isolated to confirm their chemical structures through NMR spectroscopy along with molecular modelling and MS/MS fragmentation analysis.

Compound **1** was obtained as a brown powder and its molecular formula was determined to be C_36_H_24_O_18_ from its deprotonated ion peak at *m/z* 743.0895 (calcd. for [M − H]^−^, 743.0890, Appendix A). The chemical shifts observed in the ^1^H and ^13^C NMR spectra (Appendix A), proton resonances in the downfield region of the aromatic chemical shift range, suggested a polyphenolic structure. In the ^1^H NMR spectrum, eleven aromatic protons were observed, including one proton singlet at δ_H_ 6.12 (H-3), one 2H overlapped singlet at δ_H_ 5.86 (H-3″″, 5″″), and four sets of *meta*-coupled doublets [δ_H_ 6.12 (overlap, H-5″)/5.70 (d, 1.5, H-3″), 6.09 (br s, H-6′)/5.99 (d, 1.5, H-2′), 5.97 (d, 2.7, H-8)/5.81 (d, 2.7, H-6), and 589 (d, 2.7, H-5‴)/5.54 (d, 2.7, H-3‴)]. The ^13^C NMR signals corresponded to thirteen unsubstituted (δ_C_ 92.7–100.7 ppm) and twenty-three oxygenated aromatic carbons (δ_C_ 122.1–158.1 ppm), which were assigned based on their HSQC and HMBC correlations (Appendix A). These data suggested that the compound contained six phloroglucinol units. The coupling constants of the proton signals and HMBC data further confirmed the pattern of the substituents on **1** (Appendix A). The signal at δ_H_ 6.12 (1H, s) and the *meta*-coupled doublets at δ_H_ 5.89 (d, 2.7) and 5.81 (d, 2.7) were similar to those of eckol [30], and they were thus assigned to H-3 and H-8/H-6 of rings A and B, respectively. Similarly, the structure of **1** was presumed to include the C, D, E, and F rings of fucodiphloroethol G by comparison with the reported NMR data [31]. The two carbon signals at δ_C_ 100.7 and 101.4 ppm also indicated that the structure of **1** contained one C−C bond similar to that of fucodiphloroethol G [31]. However, the asymmetric structure of the C ring revealed that fucodiphloroethol and eckol moieties of **1** were linked through the C ring. The ROESY correlation of H-3 (A ring) with 5′-hydroxyl (C ring) (Appendix A) suggested a phenyl linkage connecting the C and A rings. The long-range couplings between the proton of rings A and C, as well as rings B and E, were explained by their positions derived from three-dimensional structure optimization using molecular modelling (Appendix A). A comprehensive explanation of the structure was obtained from the MS/MS data. As shown in Appendix A, the fragment ion corresponding to C_24_H_15_O_11_ (*m/z* 479.0608, calcd 479.0614, Δ −0.6 mmu) was the main fragment of **1**, and the other detected fragments were explained by the losses of phloroglucinol units (−126 amu) or pentahydroxyl dibenzodioxin (263 amu). In particular, the fragments of **1** at *m/z* 583.0716, 479.0608, 353.0284, and 265.0342 amu were from the losses of one phloroglucinol and two water molecules (−126 −34 amu), two phloroglucinol units and one oxygen atom (−248 −16 amu), three phloroglucinol and one water (−372 –18 amu), and pentahydroxyl dibenzodioxin and two protons (−263 −2 amu). Therefore, the chemical structure of **1** was determined to be dibenzodioxin-fucodiphloroethol (DFD).

Compound **2** was obtained as a brown powder and the molecular formula of **2** was established to be C_54_H_34_O_27_ on the basis of its molecular ion in the negative mode (HPLC-qTOFMS, *m/z* 1113.1215, calcd. for [M − H]^−^, 1113.1215) (Appendix A). The ^1^H NMR spectrum of **2** (Appendix A) showed a total of sixteen proton signals, including three proton singlets at δ_H_ 6.21 (1H, s, H-5″″″), 6.14 (1H, s, H-4″″‴), and 5.92 (1H, s, H-3); two sets of *meta*-coupled doublets at δ_H_ 6.01 (1H, d, 2.2 Hz, H-6), 5.88 (1H, d, 2.2 Hz, H-8), 5.89 (1H, d, 2.4 Hz, H-5‴), and 5.55 (1H, d, 2.4 Hz, H-3‴); two sets of *meta*-coupled 2H signals at δ_H_ 5.87 (2H, s, H-3″, 5″), 5.74 (2H, d, 1.8 Hz, H-2′, 6′), and 5.81 (1H, br t, H-4′); and, two sets of *meta*-coupled singlets at δ_H_ 6.14 (1H, br s, H-5″″), 5.71 (1H, br s, H-3″″), 6.11 (1H, br s, H-5″‴), and 6.00 (1H, br s, H-3″‴). The ^13^C NMR spectrum of **2** (Appendix A) showed 54 signals in the chemical shift range of aromatic carbons. The thirteen protonated carbon signals from δ_C_ 92.7 to 98.7 ppm, along with thirty-six oxygenated aromatic carbon signals that were located in the range of 121.4–163.5 ppm, suggested the presence of nine aromatic rings in the structure of **2**. All of the carbon signals in ^13^C NMR spectrum of **2** were assigned based on their HSQC and HMBC correlations (Appendix A). When compared with the reported phlorotannins, the chemical shifts of the atoms in rings A, B, C, and D of **2** were similar to those belonging to 7-phloroeckol [32]. Similarly, the chemical shifts of the atoms in rings D, E, F, and G were assigned based on the structure of fucodiphloroethol G [31]. The C−C bond between rings F and G of **2** was confirmed from the chemical shifts at δ_C_ 100.7 and 101.5 ppm. In addition, the remaining two aromatic rings of **2** were suggested to be rings H and I based on the MS/MS fragmentation analysis of **2**. The fragment at 265 amu was suggested by the vicinal trihydroxylated structure of compound **2** (Appendix A), which was characteristic of a bifuhalol unit that contained an extra hydroxyl group on the terminal phlorotannin unit [27]. The connectivity between all of the fragments was further determined by the ROESY spectrum (Appendix A). The correlations of the signals corresponding to H-2′/6′ in ring A with H-3 (B ring) and H-6 (C ring) and between H-6 (C ring) with H-3 (D ring) confirmed the connectivity of these rings, as shown in Appendix A. The 7-phloroeckol moiety was connected to the 1‴ position of the ring E based on the ROESY correlation between the H-3″ (D ring) and H-6‴ (E ring). Similar to compound **1**, the long-range correlations of the protons on rings A and D, as well as on rings B and G of **2**, were further supported by the optimized three-dimensional structure obtained using molecular modelling (Appendix A). The fragment information from HRMS/MS data of **2,** as described in Appendix A, also supported the chemical structure of compound **2**. The fragments that were observed in Appendix A at *m/z* 989.0983, 879.0299, 556.0534, and 265.0334 amu were explained by the losses of one phloroglucinol unit (−126 amu); two phloroglucinol units and the addition of one water (−252 and +18 amu); one eckol, one bifuhalol, the addition of one acetic acid, and one water (−372 −263 +60 +18 amu); and, a bifuhalol fragment at 265.0334 amu. Finally, the exact structure of **2** was elucidated as dibenzodioxin-fucodiphloroeckol (DFE).

### 2.3. Discussion of the Potential Applications of the Developed Dereplication Strategy and Phlorotannins

In this paper, the developed dereplication strategy using RMD values is based on the general dereplication strategy that determines the similarities between the MS/MS fragmentation patterns of standard compounds and those of the unknown compounds. However, unlike the general dereplication method, RMD can be used to group compounds of the same type to show the overall phytochemical trends by the types of secondary metabolites. The extracts of *E. cava* or phlorotannins (dieckol, phloroglucinol, eckol, phlorofucofuroeckol, and 7-phloroeckol, etc.) have been reported to have antiviral activities, such as PEDV [9], viral hemorrhagic septicemia virus (VHSV) [33], human papilloma virus (HPV) [34], and murine norovirus (MNV) [35]. However, in this study, a method of producing a bioactive fraction was developed, and this fraction could be used to obtain antiviral materials from a total extract. Additionally, seven lead compounds were identified and purified from the bioactive fraction. Therefore, the fraction of the *E. cava* extract and the isolated phlorotannins have been shown to be potentially industrially applicable in the prevention or treatment of influenza A viruses.

## 3. Conclusions

The present study discussed using MS/MS fragmentation analysis coupled with RMD filtering as a dereplication method to identify phlorotannins, which are bioactive substances in the *Ecknolia* genus, especially in *Ecknolia cava*. To the best of our knowledge, this is the first attempt to apply and post-evaluate the accuracy of the RMD values in conjunction with MS/MS analysis for the identification of phlorotannins. The results of the dereplication study provide a quick initial screening for the phlorotannin motif, as these compounds show RMD values of less than 200 ppm, typically with fragments from losses of phloroglucinol units. In addition, different ranges of RMD values suggest the presence of different types of structures that are present in the compound. MS/MS was not only useful for dereplication but it also provided strong evidence to confirm the structures of phlorotannins by their fragments. Based on this dereplication method, two new compounds were identified and further elucidated by spectroscopy. In addition, this study was conducted to identify the antiviral lead compounds from *E. cava* using bio-guided isolation. Our results suggested that phlorofucofuroeckol A (**12**) from *E. cava* plays a key role in the antiviral activities of this seaweed against H1N1 and H9N2 virus. The strong inhibitory effects of this compound against neuraminidase and hemagglutinin suggested that it could be a potential agent for the further development of antiviral drugs.

## 4. Materials and Methods 

### 4.1. General Experimental Procedures

Optical rotations were determined on a JASCO P-2000 polarimeter (JASCO International Co. Ltd., Tokyo, Japan). The IR spectra were recorded on a Nicolet 6700 FT-IR spectrometer (Thermo Electron Corp., Waltham, MA, USA). NMR data were measured in DMSO-*d*_6_ and recorded on 300, 800, and 850 MHz spectrometers at the College of Pharmacy, Seoul National University, Korea. ESI-MS data were recorded on an Agilent 1100 series LC/MSD TRAP (Agilent Technologies, Waldbronn, Germany). The active fractions were analysed by TLC (thin-layer chromatography) on silica gel 60 F254 and RP-18 F254 plates from Merck (Darmstadt, Germany). Spraying with a vanillin reagent that contained 0.5 g of vanillin, 80 mL of sulfuric acid, and 20 mL of ethanol developed the plates, and the compounds on the TLC plates were detected at 254 and 365 nm. Column chromatography separations were carried out using various resins, including Diaion HP-20 (Mitsubishi Chemical Co. Tokyo, Japan), Sephadex LH-20 (Sigma-Aldrich Corp, St. Louis, Missouri, USA), silica gel (Merck, 40–63 µm particle size), and C_18_-RP silica gel (Merck, 40–63 μm particle size). Preparative HPLC separations were performed by using a Gilson system with an Optima Pak C_18_ column (10 mm × 250 mm, 10 µm in particle size, RS Tech, Seoul, Korea) and a UV detector at wavelengths of 205 and 254 nm. All of the solvents that were used for extraction and isolation were of analytical grade. The most stable configuration of compounds **1** and **2** (Appendix A) were calculated by CONFLEX 8 (Conflex Corp., Tokyo, Japan) using molecular mechanics force-field (MMFF94s) calculations with a search limit of 1.0 kcal/mol.

### 4.2. Plant Material

*E. cava* was collected from Cheongsando Island, Republic of Korea in July 2016. Prof. T. O. Cho botanically identified the plant sample. A voucher specimen (TCMBRB0019/TC9484) was deposited at the Algae Systematics laboratory of Chosun University, College of Natural Sciences, Gwang-ju, Republic of Korea.

### 4.3. Extraction and Isolation

Dried powder of *E. cava* (100 g) was extracted with 80% MeOH (5 L) at room temperature under ultra-sonication for two days. After filtration, the combined MeOH solution was concentrated under reduced pressure to yield dry extract (20.4 g). The crude extract was suspended in distilled water (2 L) and then fractionated using Diaion HP-20. Four litres of water was used to elute the salts and then two litres each of 30% MeOH, 70% MeOH, 100% MeOH, and 100% acetone were used as the elution systems to obtain five fractions. The fraction eluted with 30% MeOH was chromatographed on a reversed-phase silica gel column with a gradient of MeOH/H_2_O from 30% to 100% to yield five subfractions (F1–F5) (Appendix A). Sub-fraction F2 was subjected to a Sephadex LH-20 column that was eluted with 100% MeOH to afford four fractions (F2.1–F2.4). Fraction F2.3 was further purified on a Sephadex column with 100% MeOH to give three subfractions (F2.3.1–3). Subfraction F2.3.2 was further purified by semi-preparative HPLC [Optimapack C_18_ column (10 mm × 250 mm, 10 µm particle size, RS Tech, Seoul, Korea); mobile phase MeCN in H_2_O containing 0.1% HCO_2_H (0–40 min: 35 to 100% MeCN, 41–51 min: 100% MeCN); and, flow rate: 2 mL/min] to yield compounds **1** (6.0 mg) and **2** (5.0 mg). The 70% MeOH fraction was separated by reversed-phase chromatography (40–63 µm particle size) with a stepwise elution gradient of MeOH/H_2_O (2:3–1:0) to afford four subfractions (F70.1–F70.4). The fraction F70.1 was subjected to HPLC [Optimapack C_18_ column (10 mm × 250 mm, 10 µm particle size, RS Tech, Seoul, Korea); mobile phase MeCN in H_2_O containing 0.1% HCO_2_H (0–65 min: 20% MeCN, 65 min: 100% MeCN); flow rate: 2 mL/min] to afford compounds **11** (dieckol, 15.0 mg), **13** (dibenzo[1,4]dioxine-2,4,7,9-tetraol, 3.0 mg), **6** (eckol, 2.0 mg), and **10** (6,6′-bieckol, 2.0 mg). Subsequently, fraction F70.2 was separated by HPLC [Optimapack C_18_ column (10 mm × 250 mm, 10 µm particle size, RS Tech, Seoul, Korea); mobile phase MeCN in H_2_O containing 0.1% HCO_2_H (0–65 min: 20% MeCN, 65 min: 100% MeCN); flow rate: 2 mL/min] to give compounds **12** (phlorofucofuroeckol A, 15.0 mg) and **14** (dioxinodehydroeckol, 4.0 mg). Fraction F70.3 was further purified by HPLC (Optimapack C_18_ column (10 mm × 250 mm, 10 µm particle size, RS Tech, Seoul, Korea); mobile phase MeCN in H_2_O containing 0.1% HCO_2_H (0–60 min: 30% MeCN, 65 min: 100% MeCN); flow rate: 2 mL/min) to yield compound **16** (fucofuroeckol A, 4.0 mg). All of the isolated materials and related information were kept in the Korea Bioactive Natural Material Bank (KBNMB).

#### 4.3.1. Dibenzodioxin-fucodiphloroethol (DFD) (**1**)

Brown powder; [*α*]${}_{D}^{20}$ +1.5 (*c* 0.1, MeOH); UV(MeOH) λ_max_ (log ε) 230 (2.15); IR *ν*_max_ 3863, 3840, 3734, 3647, 3272, 2675, 2348, 1748, 1620, 1540, 1508, 1487, 1396, 1269, 1198, 1150, 1090, 1039, 1024 cm^−1^; ^1^H and ^13^C NMR data (800/200 MHz; DMSO-*d*_6_), see Appendix A; HRFABMS *m/z* 743. 0895 (calcd. for [M − H]^−^, 743.0890).

#### 4.3.2. Dibenzodioxin-fucodiphloroeckol (DFE) (**2**)

Brown powder; [*α*]${}_{D}^{20}$ −22.7 (*c* 0.1, MeOH); UV(MeOH) λ_max_ (log ε) 230 (2.15); IR *ν*_max_ 3840, 3223, 2348, 1748, 1620, 1508, 1474, 1396, 1259, 1088, 1039, 1023, 997, 827 cm^−1^; ^1^H and ^13^C NMR data (850/212.5 MHz; DMSO-*d*_6_), see Appendix A; HRFABMS *m/z* 1113.1215 (calcd. for [M − H]^−^, 1113.1215).

### 4.4. HPLC-qTOFMS Measurement 

HPLC-qTOFMS was performed on an Agilent 6500 series instrument (Agilent technologies, USA) with a dual ESI interface at the College of Pharmacy, Seoul National University, Korea. A YMC-Triart C_18_ column (150 mm × 3.0 mm i.d., 5 μm) was used for separation. The elution gradient consisted of H_2_O (A) and MeCN (B) (both buffered with 0.01% formic acid) increased from 10% B to 60% B (for fraction EC30) and 100% B (for fraction EC70) in 40 min (0.3 mL/min), where it was held in 12 min (1 mL/min), returned to 10% B in 0.1 min, and then maintained for 5 min (0.3 mL/min). Nitrogen was used as the drying gas and collision gas in the ESI source. The electrospray ion source parameters were as follows: drying gas flow rate, 10 L/min; heated capillary temperature, 350 °C; sheath gas temperature of 350 °C and flow of 12 L/min; nebulizer pressure, 30 psi; VCap, fragmentor, skimmer, and octopole RF peak voltages that were set at 4000, 180, 60, and 750 V, respectively. The detection was carried out in positive and negative electrospray ionization modes, and the spectra were recorded by MS scanning in the range of *m/z* 100–1000. The MS/MS analyses were carried out by targeted fragmentation and the collision energy was set at 20–80 eV for fractions EC30 and EC70 and at 50 eV for single compounds. MassHunter software version B.06.01 (Agilent Technology) was used to control the LC–MS/MS system. Version B.07.00 was used for data acquisition, analysis, and processing, including the prediction of chemical formula and exact mass calculation. 

### 4.5. Cell Cultures and Viruses

MDCK cells were provided by American Type Culture Collection (ATCC, Manassas, VA 20108, USA) and cultured in Dulbecco’s modified Eagle’s medium (DMEM) containing 10% FBS, 100 U/mL penicillin, and 100 μg/mL streptomycin. Influenza A viruses (H1N1 A/PR/8/34 or H9N2 A/chicken/Korea/01210/2001) were obtained from Choong Ang Vaccine Laboratory, Korea, and they were stored at −80 °C.

### 4.6. Cytotoxicity Assay

The cell viabilities were evaluated by a MTT (3-(4,5-dimethyl-2-thiazolyl)-2,5-diphenyl-2H-tetrazolium bromide) assay. Briefly, MDCK cells were seeded in 96-well plates at 1 × 10^5^ cells per well and then incubated for 24 h. The cultures were then replaced with DMEM-free serum and treated with the test compounds for 48 h. To avoid solvent toxicity, the final concentration of dimethyl sulfoxide (DMSO) was maintained at 0.05% (*v*/*v*). Subsequently, 20 μL of a 2 mg/mL MTT solution was added to each well and the plates were incubated for 4 h at 37 °C. After removing the medium, the formazan crystals were dissolved with 100 μL of DMSO and the absorbance was measured at 550 nm using a microplate reader (VersaMaxTM, Randor, PA, USA). All of the experiments were performed in triplicate. Regression analysis was applied to calculate the 50% cytotoxic concentration (CC_50_).

### 4.7. Cytopathic Effect (CPE) Assay

MDCK cells were seeded onto 96-well plates and incubated for 24 h. After washing twice with phosphate-buffered saline (PBS), the cells were infected with influenza viruses using DMEM that contained 0.15 µg/mL trypsin and 5 µg/mL BSA for 2 h. The cells were then washed with PBS, and new media containing the fractions or test compounds at different concentrations was added. After incubation for three days at 37 °C under a 5% CO_2_ atmosphere, the medium was replaced with DMEM, 20 µL of 2 mg/mL MTT was added to each well, and the plates were incubated for an additional 4 h. The next steps followed a typical cytotoxicity assay procedure and the 50% effective concentration (EC_50_) was calculated by regression analysis. 

### 4.8. Western Blot Analysis

MDCK cells were grown into 6-well plates and infected with H1N1 A/PR/8/34 virus for 2 h. The cells were then washed with PBS and the media was replaced with new media with various concentrations of the test compounds. After 24 h of incubation, the cells were washed with cold PBS and then lysed with 100 µL of lysis buffer [50 mM Tris-HCl (pH 7.6), 120 mM NaCl, 1 mM EDTA, 0.5% NP-40, and 50 mM NaF]. The supernatants were collected, and the protein concentrations were determined using a protein assay kit (Bio Rad Laboratories Inc., USA). The aliquots of the lysates were boiled for 5 min and loaded onto 10% or 12% SDS-polyacrylamide gels. The gels were then electrophoresed and electrotransferred to polyvinylidene fluoride membranes (PVDF 0.45 µm, Immobilon-P, USA). After blocking with a 5% skim milk solution, the membranes were incubated overnight at 4 °C with primary antibodies, namely, neuraminidase (Gene Tex, San Antonio, TX, USA), hemagglutinin (Sigma, St Louis, MO, USA), or mouse monoclonal actin (Abcam, Cambridge, UK). After washing a few times with PBS, the membranes were incubated with secondary antibodies for 2 h and then detected by a chemiluminescence Western blotting detection kit (Thermo Fisher Scientific., Rockford, IL, USA) using an Image Quant^TM^ LAS4000 imaging system.

### 4.9. Immunofluorescence Assay

The MDCK cells were maintained on sterile glass coverslips and then infected with H1N1 A/PR/8/34 virus for 2 h. After that, the cells were transferred to new media containing the test compounds at various concentrations. The cultures were incubated at 37 °C under a 5% CO_2_ atmosphere for 24 h. After fixation and permeabilization, the cells were blocked with 1% bovine serum albumin (BSA) solution (Sigma, St. Louis, MO, USA) for 1 h and then incubated overnight with monoclonal neuraminidase antibody (Gene Tex) that was diluted 1:50 in PBS. The cells were then incubated with the secondary antibody FITC (fluorescence isothiocyanate)-conjugated goat anti-Rb IgG (Abcam, Cambridge, UK) for 1 h, and the nuclei were stained with 500 nM 4′,6-diamidino-2-phenylindole (DAPI, Thermo Fisher Scientific, USA) solution for 5 min at room temperature. The slides were mounted with mounting medium for fluorescence (Vectashield, Vector Lab, Inc., USA). The immunofluorescence images were acquired with fluorescence microscopy (Olympus ix70 Fluorescence Microscope, Olympus Corporation, Tokyo, Japan). 

### 4.10. Statistical Analysis 

The results are presented as the mean ± SD of three independent experiments. The significant differences between the groups were calculated by one-way analysis of variance (ANOVA) using Tukey’s or Duncan’s post hoc tests, which were conducted with SPSS Statistics 23 software (SPSS, Inc., Chicago, IL, USA). Statistical significance was accepted at * *p* < 0.05, ** *p* < 0.01, and *** *p* < 0.001. ImageJ software was applied for Western blot analysis.

## Figures and Tables

**Figure 1 marinedrugs-17-00149-f001:**
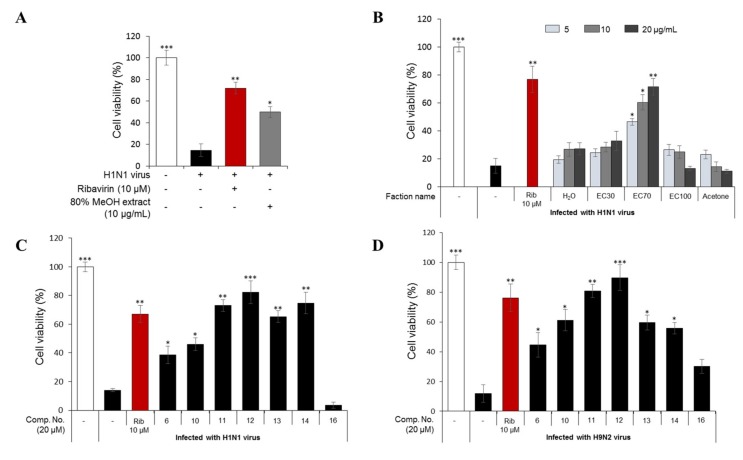
Cytopathic effect (CPE) inhibition assay to determine the antiviral activities of the 80% methanol extract of *E. cava* at 10 µg/mL (**A**) and five fractions (separated from the total extract with an open HP-20 column) (**B**) against the H1N1 A/PR/8/34 virus. In addition, the antiviral effects of compounds **6**, **10**, **11**, **12**, **13**, **14**, and **16** against H1N1 A/PR/8/34 virus (**C**) and H9N2 A/chicken/Korea/01210/2001 virus (**D**). Madin–Darby canine kidney (MDCK) cells were infected with the influenza viruses for 2 h and then treated with the test compounds or ribavirin (10 µM) as a positive control. The percentage of cell survival was evaluated after three days of incubation using a CPE inhibition assay. Data are expressed as the mean ± SD (*n* = 3), * *p* < 0.05, ** *p* < 0.01, and *** *p* < 0.001 as compared to the virus control group.

**Figure 2 marinedrugs-17-00149-f002:**
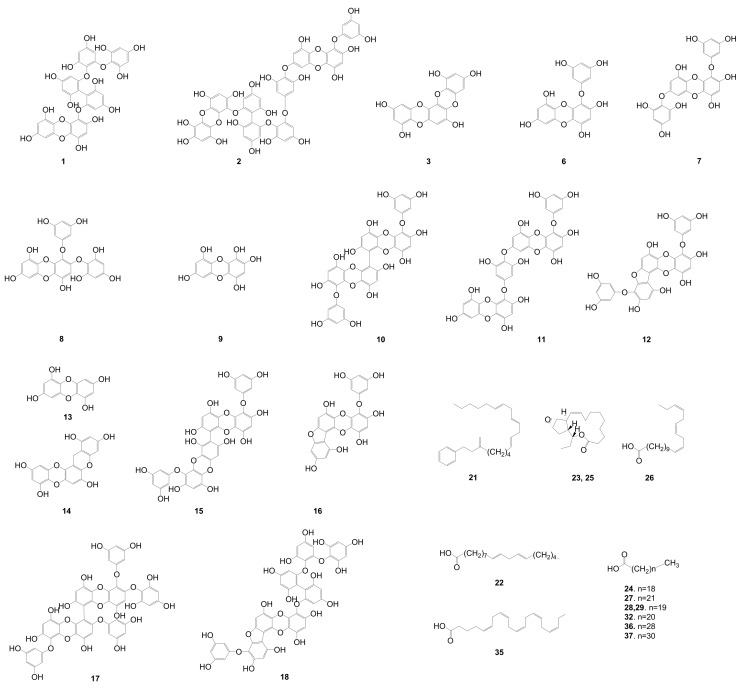
Identified phlorotannins and other compounds predicted by dereplication with high-resolution mass spectrometry and relative mass defect (RMD) values based on in-house and online databases. MassHunter software was used in this prediction.

**Figure 3 marinedrugs-17-00149-f003:**
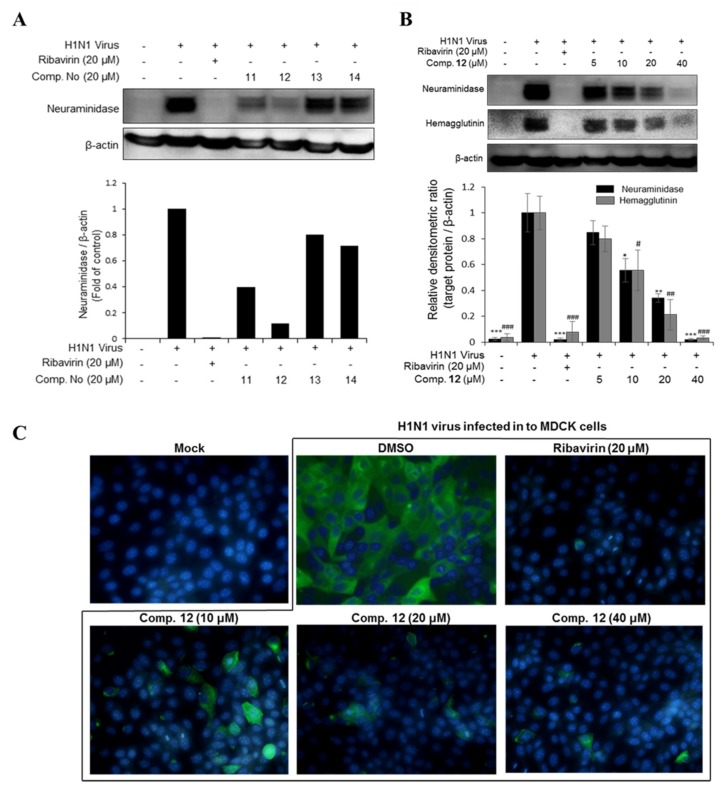
(**A**) The inhibitory effects of compounds **11**, **12**, **13,** and **14** on viral protein synthesis. (**B**) The concentration-dependent inhibitory effect of compound **1****2** on viral protein synthesis. MDCK cells were infected with influenza H1N1 virus for 2 h and then treated with the test compounds or ribavirin (20 µM) as a positive control for 24 h. Western blotting was performed with β-actin as an internal control and specific antibodies (neuraminidase and hemagglutinin). The data are presented as the mean ± SD (*n* = 2–3), * *p* < 0.05, ** *p* < 0.01, *** *p* < 0.001 and ^#^
*p* < 0.05, ^##^
*p* < 0.01, ^###^
*p* < 0.001 when compared to the neuraminidase and hemagglutinin virus control groups, respectively. (**C**) Compound **1****2** decreased neuraminidase protein expression in viral-infected cell cytoplasm at various concentrations (10, 20, and 40 µM). The fluorescence images were determined by immunocytochemistry using a fluorescence microscope.

**Figure 4 marinedrugs-17-00149-f004:**
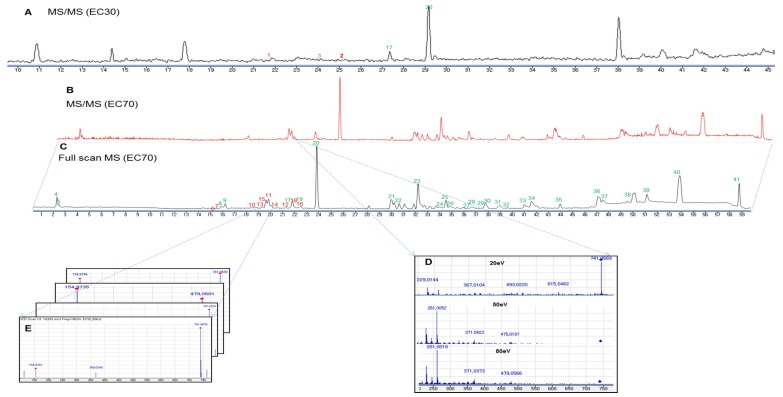
Base peak chromatograms of the high-performance liquid chromatography—quadrupole time-of-flight mass (HPLC-qTOFMS) spectra of the EC30 (**A**) and EC70 (**B**–**E**) fractions of *E. cava* in negative ionization mode at a collision energy of 50 eV. The chromatograms were generated using MassHunter software. Full scan, high-resolution mass spectrometry screening (**C**,**E**); MS/MS identification using an in-house library of the *Ecklonia* genus with different collision energies from 20 to 80 eV (**B**,**D**).

**Table 1 marinedrugs-17-00149-t001:** MS/MS fragmentation of the peaks detected in the total ion chromatograms of the EC30 and EC70 fractions of *E. cava* in negative ionization mode.

No	Compound Name	RT (min)	MS ^1^	MS ^2^	[M − H]^−^	Chemical Formula	Calcd. Mass	Dif.	RMD
**1**	Compound **1**	21.793	195, 247, 389, 479, 583, 743	139, 231, 285, 353, 447, 493, 643, 743	743.0881	C_36_H_23_O_18_	743.0884	−1.73	119
**2**	Compound **2**	25.279	195, 359, 479, 556, 663, 982, 1113	231, 353, 461, 705, 925, 1104, 1113	1113.1146	C_54_H_33_O_27_	1113.1209	−7.94	103
**3**	Benzo[1,2-b:3,4-b′]bis[1,4] benzodioxin-1,3,6,9,11-pentol	24.015	195, 211, 246, 292, 369	193, 230, 267, 285, 310, 369	369.0224	C_18_H_9_O_9_	369.0247	−4.80	61
**4**	Unknown	2.268	162, 197	163, 821	162.8392	-			5154
**5**	Unknown	2.631	122, 139, 168, 195	111, 139, 171	195.9504	-			4850
**6**	Eckol	15.543	154, 371	217, 246, 282, 371	371.0446	C_18_H_11_O_9_	371.0403	6.56	120
**7**	7-Phloroeckol	15.845	154, 495	263, 297, 387, 488, 495, 616	495.0511	C_24_H_15_O_12_	495.0564	−7.28	103
**8**	2-Phloroeckol	16.003	154, 263, 495	201, 229, 283, 346, 495	495.0528	C_24_H_15_O_12_	495.0564	−6.00	107
**9**	Dibenzo[b,e][1,4]dioxin-1,2,4,7,9-pentol	16.209	137, 155, 263	207, 218, 263	263.0157	C_12_H_7_O_7_	263.0192	−5.92	60
**10**	6,6′-Bieckol	18.854	154, 741	229, 261, 371, 479, 585, 666, 741	741.0647	C_36_H_21_O_18_	741.0728	−9.00	87
**13**	Dibenzo[1,4]dioxine-2,4,7,9-tetraol	19.274	123, 155, 247	141, 195, 247, 385, 479	247.0243	C_12_H_7_O_6_	247.0243	0.00	98
**15**	6,8′-Bieckol	19.569	741	260, 371, 479, 615, 741	741.0737	C_36_H_21_O_18_	741.0728	3.00	99
**11**	Dieckol	19.917	321, 741	229, 261, 371, 430, 545, 714, 741	741.0723	C_36_H_21_O_18_	741.0728	−2.24	98
**14**	Dioxinodehydroeckol	20.370	123, 196, 325, 369	123, 161, 173, 199, 261, 369	369.0269	C_18_H_9_O_9_	369.0247	4.69	73
**12**	Phlorofucofuroeckol A	21.796	155, 601	245, 385, 447, 493, 601	601.0640	C_30_H_17_O_14_	601.0618	4.69	106
**17**	2,7″-phloroglucinol 6,6′-bieckol (PHB)	22.018	155, 973	229, 353, 427, 493, 707, 806, 973	973.1153	C_48_H_29_O_23_	973.1100	7.28	118
**18**	974-A	22.050	113, 601, 973	229, 353, 393, 605, 707, 805, 941	973.1038	C_48_H_29_O_23_	973.1100	−7.87	107
**19**	Unknown	22.44	155, 369, 551	223, 304, 551, 583, 710	551.1816	-			329
**16**	Fucofuroeckol A	22.897	155, 477,	2551, 352, 477, 545	477.0425	C_24_H_13_O_11_	477.0457	−5.66	89
**20**	Unknown	23.799	242, 310	201, 271, 348	242.1758	-			726
**21**	6,9,12-Octadecatrienoic acid, (6Z,9Z,12Z)	29.966	277, 527	264, 353, 481,	277.2125	C_18_H_29_O_2_	277.2168	−6.56	767
**22**	9,12-Octadecadienoic acid (9Z,12Z)	30.496	279	218, 248, 279, 346, 380	279.2168	C_18_H_31_O_2_	279.2324	−12.49	776
**23**	Ecklonialactone B (R/S)	32.281	265, 291	201, 291	291.2020	C_18_H_27_O_3_	291.1960	7.75	694
**24**	Eicosanoic acid	34.206	311	225, 311, 349	311.2848	C_20_H_39_O_2_	311.2950	−10.10	915
**25**	Ecklonialactone B (S/R)	34.499	291, 555, 623	251, 291, 411,651	291.1938	C_18_H_27_O_3_	291.1960	−4.96	666
**26**	11,14,17-Eicosatrienoic acid, (11Z,14Z,17Z)-	34.771	305	211, 284, 305, 248, 583, 804	305.2410	C_20_H_33_O_2_	305.2481	−8.43	790
**27**	Tricosanoic acid	36.294	353	257, 333, 353, 529	353.3407	C_23_H_45_O_2_	353.3420	−3.61	964
**28**	Heneicosanoic acid	36.667	325	225, 239, 248, 267, 282, 325	325.3131	C_21_H_41_O_2_	325.3107	4.90	962
**29**	Unknown	37.806	293, 325	281, 325, 386, 449, 674	325.1812	-			557
**30**	Unknown	37.869	239, 293	207, 239, 243, 383	239.0709	-			297
**31**	Unknown	38.947	321	248, 321, 399, 572, 815	321.2178	-			678
**32**	Docosanoic acid	39.599	339	226, 339, 433, 660, 809	339.3292	C_22_H_43_O_2_	339.3263	5.39	970
**33**	Unknown	41.028	346	206, 254, 330, 346, 642	346.1092	-			316
**34**	Unknown	41.59	346, 485	280, 421, 485	485.2671	-			550
**35**	5,8,11,14,17-Eicosapentaenoic acid, (5Z,8Z,11Z,14Z,17Z)	43.93	301, 369	205, 269, 301, 440	301.2099	C_20_H_29_O_2_	301.2168	−8.31	697
**36**	Triacontanoic acid	47.116	346, 451, 535, 691, 775	295, 387, 451, 456	451.4499	C_30_H_59_O_2_	451.4515	−4.00	997
**37**	Dotricontanoic acid	47.575	346, 479, 609, 691, 775	461, 479, 648, 866	479.2903	C_32_H_63_O_2_	479.4828	−5.00	606
**38**	Unknown	49.963	223, 297, 441, 535, 701, 849	254, 333, 441, 673	441.2074	-			470
**39**	Unknown	51.118	149, 223	149, 448	149.0021	-			14
**40**	Unknown	53.856	149, 223, 297	149, 221, 350, 630.7708	149.0086	-			58
**41**	Unknown	58.796	135	135.9698, 287.5036	135.9751	-			7171

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
