# Peer review of "Dereplication by High-Performance Liquid Chromatography (HPLC) with Quadrupole-Time-of-Flight Mass Spectroscopy (qTOF-MS) and Antiviral Activities of Phlorotannins from Ecklonia cava"

_marinedrugs, 2019, doi:10.3390/md17030149_

Round 1
Reviewer 1 Report
A general observation is that the study contains a lot of new and interesting information on the antiviral activities of a number of new phlorotannins characterized in Ecklonia cava algae using dereplication by HPLC-QTOF-MS, but needs a much better structure and re-allocation of tables and figures between the main manuscript text and the supplementary information, in order to make the manuscript more comprehensive and to better focus on the significant findings of the study. The almost complete lack of a Discussion section is also noted (despite the fact that section 2 is considered as “Results and Discussion”), which needs to be definitely added. There is also a general need to improve the use of English language, so the authors are kindly requested have their manuscript checked by a native English speaker or a relevant professional service. Detailed comments follow:
Abstract:
- Page 1, line 17: Please explain PEDV at this first instance in the text.
1. Introduction:
- Page 2, lines 83-92: It is common to provide the purpose of the study at the end of the introduction section. However, this paragraph is actually a summary of purpose, methodology, results and conclusions. This should be avoided in this part; it either belongs to the abstract or the conclusions’ sections but certainly not to the introductory one.
2. Results and Discussion:
- As noted above, discussion of results and comparison to the findings of other works and/or to current knowledge on the researched topic is practically missing, as well as discussion on the significance and impact of the study results. The authors should therefore add a discussion part in all relevant subsections (e.g. compare to already characterised phlorotannins, compare to current knowledge on the antiviral activity of these compounds, discuss the potential applications of their findings, etc.). Alternatively, a separate discussion section can be added, to the authors’ convenience.
- There is inconsistency in the abbreviations used with regard to the LC-MS technology used in the study (throughout the text, not only in this section). There are actually instances of HPLC-qTOFMS, HRMS/MS, HRESIMS/MS, HRESI-MS/MS and HR-ESI-MS/MS, which practically describe the same methodology with a different terminology (e.g. page 3, lines 104-109, where 3 different abbreviations are used within 6 lines of text). The authors should choose the most appropriate abbreviation that best reflects their methodology and use it throughout the manuscript.
2.1. Bio-guided isolation and dereplication of antivial active fraction
- Page 3, line 94: Please correct “antivial” to “antiviral”
2.1.1. Bioactivity-guided fractionation with antiviral activity against human influenza H1N1 A/PR/8/34 virus
- Figure S17 contains important results and should be moved to the main section.
2.1.2. Dereplication and isolation of phlorotannins from EC70 fraction using HPLC-qTOFMS and RMD
- Page 4, line 146: Please explain what UHPLC-QQQ-MS stands for.
- Figure 3: the legend is incomplete, some information is missing (…were measured in negative mode at collision energy???).
- Figure 3 could be moved to the supplementary section.
- Some of the figures in the supplementary section are not referenced in the text (e.g. S14, S16). The authors should make sure that all figures included are used in the text, or remove accordingly.
2.1.3. Antiviral activities against H1N1 and H9N2 of major phlorotannins in the EC70 active fraction.
- Figures S18.A and B, S19 and S20 contain important results and should be moved to the main section.
2.2. Identification of two new phlorotannins from the fracion EC30 of E. cava using dereplication and RMD rules
- Page 5, line 202: Please correct “fracion” to “fraction”
- Figure 5 and (possibly) Table 2 should be moved to the supplementary section.
3. Materials and Methods:
3.3. Extraction and isolation
- Page 15, lines 341-343 and lines 346-347: please rephrase, the syntax is quite chaotic and is hard to follow. It is suggested that a tabular or a schematic approach is used.
- Pages 16-17, lines 364-435: these data would fit much better in the supplemental section (together with Table 2).
3.5. Cell cultures and viruses
- Page 18, line 463: MDCK cells are not defined (please refer to line 195 and explain the abbreviation there). Also, the choice of this cell-line vs. any other else should be justified (why was a kidney cell-line chosen for this particular research?)
3.9. Immunofluorescence assay
- Page 19, line 502: Please explain the abbreviation DAPI.
5. Conclusions
The paragraph should be numbered as 4. It is also suggested that it is placed before the Materials and Methods section, after the Results and Discussion, where it fits better.
Author contributions:
This section should describe the contribution of each of the authors to this work, both in the research part and their particular role in the manuscript’s preparation. The equal contribution of the two first authors is clear in the front matter.
Author Response
Abstract:
- Page 1, line 17: Please explain PEDV at this first instance in the text.
Answer: Based on the reviewer’s comment, we have added a brief description of PEDV to the manuscript. Porcine epidemic diarrhea virus (PEDV), which has a high mortality rate in piglets. This PEDV has been described in more detail in the introduction part. Briefly, PEDV is an enveloped single-stranded RNA virus belonging to the family Coronaviridae. It is the causative agent of porcine epidemic diarrhea, vomiting, dehydration, and high mortality in the piglets.
1. Introduction:
- Page 2, lines 83-92: It is common to provide the purpose of the study at the end of the introduction section. However, this paragraph is actually a summary of purpose, methodology, results and conclusions. This should be avoided in this part; it either belongs to the abstract or conclusions’ sections but certainly not to the introductory one.
Answer: Thank you for the reviewer’s comments, the paragraph in the introduction section has been removed in our revised manuscript.
2. Results and Discussion:
- As noted above, discussion of results and comparison to the finding of other works and/or to current knowledge on the researched topic is practically missing, as well as discussion on the significance and impact of the study results. The authors should therefore add a discussion part in all relevant subsections (e.g. compare to already characterized phlorotannins, compare to current knowledge on the antiviral activity of these compounds, discuss the potential applications of their findings, etc.). Alternatively, a separated discussion section can be added, to the author’s convenience.
Answer: We are very grateful to the reviewer’s comment. The manuscript was rewritten with the focus on the significant findings of the research and the discussion section was separated to make the explanation clearer.
Revised part: The separated discussion section; 2.3 Discussion of the potential application of the developed dereplication strategy and phlorotannins.
- There is inconsistency in the abbreviations used with regard to the LC-MS technology used in the study (throughout the text, not only in this section). There are actually instances of HPLC-qTOFMS, HRMS/MS, HRESIMS/MS, HRESI-MS/MS and HRESIMS/MS, which practically describe the same methodology with a different terminology (e.g. page 3, line 104-109, where 3 different abbreviations are used within 6 lines of text). The authors should chose the most appropriate abbreviation that best reflects their methodology and use it throughout the manuscript.
Answer: Based on the reviewer’s comment, we abbreviated the methodology in the same manner throughout the manuscript.
2.1 Bio-guided isolation and dereplication of antivial active fraction.
- Page 3, line 94: please correct “antivial” to “antiviral”
Answer: Thank you for the reviewer’s comment. Our manuscript has been corrected.
2.1.1 Bioactivity-guided fractionation with antiviral activity against human influenza H1N1 A.PR/8/32 virus
- Figure S17 contains important results and should be moved to the main section
Answer: Figure S17 was moved to Figure 3A and 3B in the revised manuscript.
2.1.2 Dereplication and isolation of phlorotannins from EC70 fraction using HPLC-qTOFMS and RMD
- Page 4, line 146: please explain what UHPLC-QQQ-MS stands for.
Answer: We explained the abbreviation of UHPLC-QqQ-MS in the revised manuscript.
UHPLC-QqQ-MS; ultrahigh-performance liquid chromatography triple quadrupole tandem mass spectrometry
- Figure 3: the legend is incomplete, some information is missing (…measured in negative mode at collision energy???)
Answer: the collision energy was already written in the previous manuscript.
(e.g HPLC-qTOFMS spectroscopic data of corresponding single compounds were measured in the negative mode at collision energy of 50 eV.)
- Figure 3 could be moved to the supplementary section
Answer: Figure 3 was moved to Figure S17 in the revised supplementary data.
- Some of the figures in the supplementary section are not referenced in the text (e.g. S14, S16). The authors should make sure that all figures included are used in the text, or remove accordingly.
Answer: Thank you for the reviewer’s comment. The supplementary data were referenced in the text appropriately in the revised manuscript.
2.1.3 Antiviral activities against H1N1 and H9N2 of major phlorotannins in the EC70 active fraction.
- Figures S18.A and B, S19 and S20 contain important results and should be moved to the main section.
Answer: Figure S18 and S19 were moved to Figure 3 and Figure 4A in the revised manuscript, respectively. However, Figure S20 is original uncropped data to show the accuracy and reliability of results. Thus, we didn’t move from the supplementary data to the manuscript.
2.2 Identification of two new phlorotannins from the fraction EC30 of E.cava using dereplication and RMS rules
- Page5, line 202: Please correct “fracion” to “fraction”
Answer: Thank you for the reviewer’s comment. The manuscript has been corrected.
- Figure 5 and (possibly) Table 2 should be moved to the supplementary
Answer: Figure 5 and Table 2 were moved to Figure S20 and Table S3 in the revised supplementary data, respectively.
3. Materials and Methods;
3.3 Extraction and isolation
- Page 15, lines 341-343 and lines 346-347: please rephrase, the syntax is quite chaotic and is hard to follow. It is suggested that a tabular or a schematic approach is used.
Answer: Thank you for the reviewer’s comment. The extraction and isolation of natural products were described according to the general explanation. However, an additional scheme (Scheme S1) for the isolation process was added to the supplementary data by followed the reviewer’s critical suggestion.
- Page 16-17, lines 364-435: these data would fit much better in the supplemental section (together with Table 2)
Answer: Identification of isolated known compounds and Table 2 were moved to the revised supplementary data, respectively.
3.5 Cell cultures and viruses
- Page 18, line 463: MDCK cells are not defined (please refer to line 195 and explain the abbreviation there). Also, the choice of this cell-line vs. any other else should be justified (why was a kidney cell-line chosen for this particular research?)
Answer: Based on the reviewer’s comments, the MDCK cell abbreviation is explained in the manuscript.
Until recently, two continuous cell lines have been approved by regulatory authorities to be used for the production of vaccines: Madin Darby canine kidney (MDCK) cells (Influenza virus) and African green monkey kidney-derived Vero cells (coronavirus such as PEDV etc.). Moreover, previous studies have shown that MDCK cells produce HA antigens that are suitable for influenza, therefore, have been used in this study. We attach below a previously reported study using MDCK cells in influenza virus strains.
Doroshenko, Alexander, and Scott A. Halperin. "Trivalent MDCK cell culture-derived influenza vaccine Optaflu®(Novartis Vaccines)." Expert review of vaccines 8.6 (2009): 679-688.
Shin, Duckhyang, et al. "Comparison of immunogenicity of cell-and egg-passaged viruses for manufacturing MDCK cell culture-based influenza vaccines." Virus research 204 (2015): 40-46.
Huh, Jungmoo, et al. "C-methylated flavonoid glycosides from Pentarhizidium orientale rhizomes and their inhibitory effects on the H1N1 influenza virus." Journal of natural products 80.10 (2017): 2818-2824.
3.9 Immunofluorescence assay
- Page 19, line 502: please explain the abbreviation DAPI.
Answer: Based on the reviewer’s comments, DAPI abbreviation is explained in the manuscript.
(DAPI : 4’,6-diamidino-2-phenylindole)
5. Conclusions
- The paragraph should be numbered as 4. It is also suggested that it is placed before the Materials and Methods section, after the Results and Discussion, where it fits better.
Answer: Based on the reviewer’s comments, the paragraph was replaced before the Materials and Methods section and after the Results and Discussion section. The Result and Discussion section was numbered 2, the conclusion section numbered 3, and Materials and Methods section numbered 4.
6. Author contributions
- This section should describe the contribution of each of the authors to this work, both in the research part and their particular role in the manuscript’s preparation. The equal contribution of the two first authors is clear in the front matter.
Answer: Thank you for reviewer’s comment. The contribution of each of the authors to this work were added in the manuscript.

Reviewer 2 Report
Improved clarity throughout would be helpful for readers.
Author Response
Improved clarity throughout would be helpful for readers.
Answer: Thank you for the reviewer’s comment. We have tried to reflect the opinions of the reviewer. We apologize for the errors in English and the formatting of our manuscript. We have sent our manuscript to American Journal of Experts (AJE, http://www.aje.com/) to make corrections and attached a certificate verifying that AJE has further corrected the English problems.

Reviewer 3 Report
The experiments are well-designed and the amount of work reported is quite impressive. However, I would suggest authors to carefully re-read the entire manuscript, because most sentences contain typing and grammatical errors, and this negatively affect the quality of the paper.
The following are minor suggestions for further improving, in my opinion, the quality of the paper:
1) Line 17: please define PED.
2) Line 19: please define H1N1.
3) Line 121: is the low RMD value a specific characteristic of phlorotannins or more in general of polyphenolic compounds? Authors should discuss about this point.
4) Figure 1: what about compounds 4 and 5?.
5) Figure 4B: what does “merge” stand for in the mock sample?
6) Line 330: “Figure 6” should be “Figure 5”.
Author Response
The following are minor suggestions for further improving, in my opinion, the quality of the paper:
- Line 17: please define PEDV
Answer: Based on the reviewer’s comments, PEDV abbreviation is explained in the manuscript.
(PEDV: Porcine epidemic diarrhea virus)
- Line 19: please define H1N1
Answer: H1N1 viral strain is one of the representative swine-origin influenza virus strains.
- Line 121: is the low RMD value a specific characteristic of phlorotannins or more in general of polyphenolic compounds? Authors should discuss about this point
Answer: Thank you for your sharp comment. Classification by the RMD values is suitable for other types of polyphenolic compounds, which corresponds to the low RMD values and can provide a wide application in screening natural resources. It is because polyphenolic compounds have a high content of oxygen atom (− 5.09 mDa), unlike other types of compounds. Polyphenol-type compounds especially exhibit absolute low mass defects.
- Figure 1: what about compounds 4 and 5?
Answer: Thank you for reviewer’s comment. The structures of compounds 4 and 5 could not be drawn in Figure 1 because they are unknown compounds.
- Figure 4B: what does “merge” stand for in the mock sample?
Answer: Thank you for reviewer’s comment. I apologize for the error. We removed the word “merge” in Figure 4B.
- Line 330: “Figure 6” should be “Figure 5”
Answer: Thank you for reviewer’s comment. Our manuscript has been corrected.

Round 2
Reviewer 1 Report
Page 7, lines 298-299 (pdf file): Please add some detail and references to the sentence "Moreover, only five phlorotannins have been reported to have antiviral activities."
Author Response
Answer: Based on the reviewer’s comments, we have changed the more specific sentence. The extract of E. cava or phlorotannins (dieckol, phloroglucinol, eckol, phlorofucofuroeckol, and 7-phloroeckol, etc.) have been reported to have antiviral activities such as PEDV [9], viral hemorrhagic septicemia virus (VHSV) [33], human papilloma virus (HPV) [34], and murine norovirus (MNV) [35].
33. Yang, H.K.; Jung, M.H.; Avunje, S.; Nikapitiya, C.; Kang, S.Y.; Ryu, Y.B.; Lee, W.S.; Jung, S.J. Efficacy of algal Ecklonia cava extract against viral hemorrhagic septicemia virus (VHSV). Fish Shellfish Immunol. 2018, 72, 273−281.
34. Kim, E,B.; Kwak, J.H. Antiviral phlorotannin from Eisenia bicyclis against human papilloma virus in vitro. Planta Med. 2015, 81, 22.
35. Eom, S.H,; Moon, S.Y.; Lee, D.S.; Kim, H.J.; Park, K.; Lee, E.W.; Kim. T.H.; Chung, Y.H.; Lee, M.S.; Kim, Y.M. In vitro antiviral activity of dieckol and phlorofucofuroeckol-A isolated from edible brown alga Eisenia bicyclis against murine norovirus. Algae. 2015, 30(3), 241−246.
Some researchers have studied antiviral activity against viral hemorrhagic septicemia virus (VHSV) with the extract of E. cava. Other researchers have reported other antiviral activities, such as human papilloma virus (HPV) or murine norovirus (MNV), with some phlorotannins from the genus Ecklonia. However, as in the present study, no studies have been proposed to isolate and standardize the whole phlorotannins.
